# The Relationship between Androgen Receptor Gene Polymorphism, Aggression and Social Status in Young Men and Women

**DOI:** 10.3390/bs12020042

**Published:** 2022-02-10

**Authors:** Nohelia T. Valenzuela, Irene Ruiz-Pérez, Carlos Rodríguez-Sickert, Pablo Polo, José Antonio Muñoz-Reyes, Ali Yeste-Lizán, Miguel Pita

**Affiliations:** 1Laboratorio de Comportamiento Animal y Humano, Centro de Investigación en Complejidad Social, Facultad de Gobierno, Universidad del Desarrollo, Las Condes 7590943, Chile; ntvalenzuela@udd.cl (N.T.V.); carlosrodriguez@udd.cl (C.R.-S.); p.polo@udd.cl (P.P.); ja.munoz@udd.cl (J.A.M.-R.); 2Departamento de Biología, Facultad de Ciencias, Universidad Autónoma de Madrid, 28049 Madrid, Spain; ireneruiz96@hotmail.com (I.R.-P.); ali.yeste.94@gmail.com (A.Y.-L.)

**Keywords:** androgen receptor, *AR* gene, aggression, social status

## Abstract

In both sexes, aggression has been described as a critical trait to acquire social status. Still, almost uniquely in men, the link between aggressiveness and the genetic background of testosterone sensitivity measured from the polymorphism in the androgen receptor (*AR*) gene has been previously investigated. We assessed the relevance of the *AR* gene to understand aggression and how aggressiveness affects social status in a cross-sectional study of 195 participants, for the first time in both young men and women. We estimated polymorphism sequences from saliva and measured aggression and self-perceived social status. Unfortunately, the results did not support our prediction because we did not find any of the expected relationships. Therefore, the results suggest that the genetic association between aggressive mechanisms and polymorphism of the *AR* gene is less straightforward than expected, at least in men, and seems to indicate that aggression is not usually used to gain social status in our population.

## 1. Introduction

Human aggression is a complex and multilevel behavioral phenomenon, in which sexual differences are present [1]. In this sense, men are more prone than women to employ physical aggression [1,2]. In contrast, women tend to rely on indirect forms of aggression, such as the use of hostile repertoires to generate social exclusion and damage to reputation (e.g., spreading rumors about a rival’s sexual behavior) [3,4]. From an evolutionary framework, sex differences in aggression can be explained using a functional approach. In this context, aggression is a behavioral trait that can be used to compete for mates and retain them, becoming thus a powerful tool specialized for deterring rivals in an intrasexual competition scenario. The 1980s seminal study of Wilson & Daly [5] demonstrates the preponderance of men to participate in risky and violent behavior. This phenomenon, known as the *young male syndrome*, is argued to result from sexual selection. It leads to a more intense aggressive physical competence in men, the sex that presents the higher reproductive variance [6,7]. Noteworthily, women do also compete with each other to secure and maintain reproductive benefits [8], such as mating access [4,9] and couple retention [10]. However, women tend to use indirect forms of aggression [1,3,11,12], probably because women have more to lose than men from physical damage derived from the use of physical aggression [1,13]. Accordingly, both sexes use aggression to compete, but through different behavioral mechanisms, to increase social status to access and retain mates.

Social status entails a privileged social position that provides access to limited resources as high-quality reproductive mates [14]. Moreover, social status is positively related to physical, cognitive, and emotional health and longevity for individuals and their families [15]. Therefore, relevant arguments explain men’s and women’s orientation to attain high social status [16,17,18]. However, men and women differ in their mechanisms to achieve social status, with men being more prone to use contest competition and women more oriented to scramble and subtle competitive tactics [18]. The difference in both mechanisms is linked to sex differences in aggression, with men forming coalitions to directly compete with and subdue rivals [19] and women using solitary and indirect forms of aggression (i.e., relational aggression) to damage rivals and gain status [11]. In this regard, the hormone testosterone has been proposed as a relevant candidate to explain sex differences in aggression (e.g., [20,21]) that determine sex-specific mechanisms to seek status.

Testosterone is a hormone produced in Leydig cells of the testes of men and, in smaller quantities, in the ovaries of women and the adrenal cortex. Its effect has implications in the ontogeny of individuals as the development and maintenance of secondary sexual characteristics [22]. However, the effect of testosterone goes beyond the development of physical attributes. For example, testosterone facilitates aggression in animals [23,24,25], and, since the 1980s, it has been considered a social hormone that influences behavior in situations linked to the maintenance or enhancement of social status [26]. However, the principal evidence of a direct relationship between aggression and testosterone came from studies in men [27], being less investigated in women (see [27,28]). The studies about genetic sensitivity to testosterone have opened a new research avenue. The effect of testosterone on human behavior is interpreted in line with the physiological capacity that testosterone has to produce a cellular chain of reactions from the activation of intracellular and membrane-bound androgen receptors involved in the implementation of aggressive behavior [29]. Accordingly, polymorphism in the androgen receptor gene is one of the main candidates to understand the variability of its expression in men [30,31].

Worldwide, research on the genetic basis of aggressive human behavior has provided encouraging results [32]. In this regard, there are currently at least 38 candidate genes proposed to be associated with aggression [33]. However, one of the most exciting open questions is the role exerted by the gene encoding for the androgen receptor (*AR*) on such behavior (e.g., [34,35,36]). This line is prolific and has significantly benefited from the clinical interest in the *AR* gene, since it has been linked to the occurrence of different diseases in men and women, such as prostate cancer (e.g., [37]), infertility [38], breast cancer [39,40], and polycystic ovary syndrome [41,42]. The *AR* gene is hosted on the X chromosome and shows polymorphism in the population that results in different androgen receptor variants [36,43] with different efficacies in their transactivating signaling. This polymorphism is characterized by the presence of an expansion of CAG triplets in the coding region, which presents variations in the population (between 8 and 35 triplets) [44]. Carriers of *AR* gene alleles with a low number of CAG repeats show greater sensitivity to testosterone and a greater transactivating capacity. Therefore, they elicit a more significant response for a specific testosterone level [43], i.e., the androgen receptors exert transcriptional control of androgen-dependent genes by binding specific regulatory sequences in the DNA [45]. In contrast, alleles with a high CAG number correspond to receptors with a more remarkable insensitivity to testosterone [43] and subsequent poor transcriptional activity. In men, it has been proposed that those carrying testosterone-sensitive alleles tend to display aggressive behaviors more intensely (e.g., [30,31]; but see [46]) and be more impulsive [47] than those carrying testosterone-insensitive alleles. There are no studies in this line for women. New studies are necessary, especially those integrating relevant social variables as social status performed in both sexes.

In the present study, we assess the relevance of the polymorphism in the androgen receptor gene to understand aggression in young men and women and how these patterns of aggression affect social status. We expect that:Testosterone has been proposed as a relevant hormone to understand aggression in interactions to gain social status in men. In this sense, the *AR* gene is a key part of this chain of reactions because of its transactivating capacity for testosterone. Therefore, we expect a negative relationship between CAG repeats in the *AR* gene and aggressiveness, especially for physical aggression. In turn, we expect a positive relationship between physical aggression and social status since this type of aggression is a promotor for contest competition for social status in men.There are no studies on this field that include women and investigate the possible effect of genetic sensibility to testosterone in both sexes over the relationship between different forms of aggression and social status; therefore, our prediction will be more speculative than for men. However, we expect a negative relationship between CAG repeats and aggression in women. In turn, we expect a positive relationship between hostility and social status since it is a typical predictor of aggression used in scramble competition for social status in women.

## 2. Material and Methods

Participants: Our sample was composed of 195 young college students from Spain, of which N = 103 were women and N = 92 were men. Participants were recruited from advertisements placed at the university campus, which provided us with a cross-sectional sample. The ethics committee of the Universidad Autónoma de Madrid (CEI 74-1343) approved the study. Each participant gave their signed consent before data collection. The study did not give any monetary or academic benefit.

Buss & Perry Aggression Questionnaire (BPAQ): We applied a Spanish adaptation [48] of the BPAQ [49]. The questionnaire has been widely employed, including in studies about the *AR* gene (see [30]). This questionnaire comprises 29 items measuring different forms of aggression: physical aggression, verbal aggression, hostility, and anger. The answers are given on a five-point Likert scale, i.e., 1 = “uncharacteristic of me” to 5 = “very characteristic of me”. Cronbach’s alphas for BPAQ obtained in the present study were similar to those achieved in Santisteban and Alvarado’s [48] (PA: 0.81 vs. 0.80; VA: 0.79 vs. 0.73; A; 0.79 vs. 0.65; H: 0.73 vs. 0.66).

MacArthur Scale for Social Status: We adapted the MacArthur Scale [50,51]. The scale consists of a self-reported measurement represented through a ladder with ten steps. Participants indicate their rank relative to others in their society and in their local social environment in terms of standard indicators of socioeconomic status, including education, income, and occupational status. Therefore, people with the highest social status are at the top of the ladder with better jobs, more money, and more education. In contrast, people at the bottom of the ladder possess worse jobs (or are unemployed), are poorer, and received less education. We employed the local measure of the scale to capture the social position in which the individuals place themselves during their everyday activities.

Estimation of the Polymorphisms in the Androgen Receptor (*AR*) gene: genotyping and CAG repeats:

The participants gave a passive saliva sample of 1 mL, through the SalivaBio Collection Aid (SCA)—(50/pk) (Salimetrics, State College, PA, USA) into a polypropylene vial (cryovial 2 mL) (Salimetrics, State College, PA, USA). The DNA of each participant was isolated using the extraction kit QIAmp DNA Mini kit (Qiagen, Hilden, Germany).

To analyze the fragments of CAG repeats of the *AR* gene, we used PCR with primers forward 5′-FAM-TCCAGAGCGTGCGCGAAGTGAT-3′ and reverse 5′-CGACTGCGGCTGTGAAGGTTG-3′ [52]. The PCR reaction was performed in a final volume of 25 ul in a reaction mixture containing about 100 ng of DNA, as well as 2 mM MgCl_2_ (Bioline, London, UK), 200 μM dNTPs (Biotools, Madrid, Spain), 10 pmol of each of the primers, 1.25 U of Taq Polymerase in 1X buffer (Bioline, London, UK) and the remaining volume of H_2_Omq. The PCR reactions were carried out in a thermal cycler (Techne TC-512, Stone, UK) under the following program: initial denaturation for 5 min at 94 °C; 35 cycles of 45 s at 94 °C, 45 s at 62 °C and 45 s at 72 °C; final extension of 10 min at 72 °C. To determine the quality of the PCR products, 1% agarose gels were run in 1X TAE (Tris-acetate-EDTA) and revealed with 1X Red Gel (Biotium, Fremont, CA, USA). The PCR products (fragments of the amplified alleles) were sequenced through capillary electrophoresis in an ABI PRISM 3100 Genetic Analyzer (Applied Biosystems, Waltham, MA, USA). Sequencing delivered a “.fsa” file associated with each sample that was analyzed using the Peak Scanner Software v1.0 (Applied Biosystems, Waltham, MA, USA). This program provides a graph showing the size of the sequences (bp) versus their weight (Da); when selecting the peaks between 200 and 300 bp, the corresponding CAG repeats’ lengths are obtained. To estimate CAG repeats in women, we used the average number of CAG sequences from both X chromosomes. Three researchers obtained the measurements independently, and the samples in which there was no absolute coincidence between them were discarded.

### Statistical Analyses

We employed a *t*-test of independent samples to analyze sex differences in the variables used in our study as a part of our descriptive statistics. We fitted two mediation analyses to test our predictions, one for men and the second one for women. In both models, social status was considered the dependent variable, the number of CAG repeats in the *AR* gene was the independent variable, and physical aggression, verbal aggression, anger, and hostility were considered multiple mediators. In this way, we can test the effects of the number of CAG repeats in the *AR* gene on the different types of aggression and if physical aggression in men and hostility in women are, in turn, employed to reach a better social status. In both models, we controlled for age. In addition to these two models, we fitted two alternative models, one for men and the other for women, considering all the types of aggression together. We calculated the minimum effect size detectable with our sample sizes in multiple linear regression involving the relationships between number of CAG repeats and aggression, between aggression and social status, and between number of CAG repeats and social status. Both for men and women, our samples allowed us to detect medium-small effect sizes (women: f^2^ = 0.080; men: f^2^ = 0.070) according to Cohen’s [53] guidelines and considering an α = 0.05 and power of 0.80. T-tests were performed with IBM SPSS 25 software (Armonk, NY, USA). Mediation analyses were performed using linear regressions and a bootstrapping method (5000 bootstraps and p = 95%) to estimate the significance of indirect effect with PROCESS macro for SPSS [54]. Finally, we employed G*Power (Version 3.1.9.6) (Düsseldorf, Germany) to calculate the effect size detectable with our sample sizes. All the analyses were two-tailed, and the level of significance was set at α = 0.05.

## 3. Results

Descriptive statistics for all the variables employed in this study and *t*-test reporting sex differences are shown in Table 1.

Regarding our first hypothesis concerning men, we found that the number of CAG repeats was not related to any type of aggression (PA: β = 0.132, t = 1.264, *p* = 0.209; VA: β = 0.084, t = 0.814, *p* = 0.418; A: β = 0.027, t = 0.264, *p* = 0.793; H: β = 0.053, t = 0.504, *p* = 0.615; Figure 1), nor to overall aggression (β = 0.105, t = 1.024, *p* = 0.309). In turn, verbal aggression was negatively related to social status (β = −0.333, t = −2.875, *p* = 0.005; Figure 1) but physical aggression, anger and hostility were not related to social status (PA: β = −0.104, t = −0.853, *p* = 0.396; A: β = 0.216, t = 1.655, *p* = 0.102; H: β = −0.063, t = −0.563, *p* = 0.575; Figure 1) as well as overall aggression (β = −0.159, t = −1.466, *p* = 0.146). In accordance with these results, the number of CAG repeats was not related neither directly to social status (β = −0.038, t = −0.369, *p* = 0.713) nor indirectly through the different types of aggression (PA: β = −0.014, Bootstrapped SE = 0.043, 95% bootstrapped CI [−0.069, 0.028]; VA: β = −0.028, Bootstrapped SE = 0.033, 95% bootstrapped CI [−0.101, 0.031]; A: β = 0.006, Bootstrapped SE = 0.024, 95% bootstrapped CI [−0.046, 0.054]; H: β = −0.003, Bootstrapped SE = 0.014, 95% bootstrapped CI [−0.037, 0.024])^6^ or through overall aggression (β = −0.017, Bootstrapped SE = 0.019, 95% bootstrapped CI [−0.063, 0.014]).

Regarding our second hypothesis concerning women, we found that the number of CAG repeats was not related to any type of aggression (PA: β = 0.015, t = 0.153, *p* = 0.879; VA: β = −0.038, t = −0.380, *p* = 0.705; A: β = 0.081, t = 0.813, *p* = 0.418; H: β = −0.031, t = −0.307, *p* = 0.760; Figure 2) nor to overall aggression (β = 0.012, t = 0.123, *p* = 0.902). In turn, none of the types of aggression were related to social status (PA: β = −0.193, t = −1.507, *p* = 0.135; VA: β = 0.093, t = 0.742, *p* = 0.460; A: β = 0.144, t = 0.998, *p* = 0.321; H: β = −0.100, t = −0.884, *p* = 0.379; Figure 2) as well as overall aggression (β = −0.037, t = −0.365, *p* = 0.716). In accordance with these results, the number of CAG repeats was not related neither directly to social status (β = −0.078, t = −0.779, *p* = 0.438) nor indirectly through the different types of aggression (PA: β = −0.003, Bootstrapped SE = 0.023, 95% bootstrapped CI [−0.049, 0.050]; VA: β = −0.004, Bootstrapped SE = 0.019, 95% bootstrapped CI [−0.048, 0.035]; A: β = 0.012, Bootstrapped SE = 0.027, 95% bootstrapped CI [−0.031, 0.079]; H: β = −0.003, Bootstrapped SE = 0.016, 95% bootstrapped CI [−0.028, 0.044]) or through overall aggression (β = −0.000, Bootstrapped SE = 0.011, 95% bootstrapped CI [−0.022, 0.026]).

## 4. Discussion

In the present study, we assessed the relationship between CAG repeats of the *AR* gene, aggression and social status in young men and women. We proposed two different predictions, one for each sex, focused on the relationship of CAG repeats of *AR* gene over specific mechanisms of aggression that differently conducted the challenge for status in men and women. However, our predictions were not supported as we did not find any expected relationships according to our theoretical proposition. Consequently, these results suggest that the genetic association between aggressive mechanisms and the polymorphism of the *AR* gene is less straightforward than expected, at least in men, and seems to indicate that aggression is not a commonly used mechanism to gain social status in our population.

The first prediction was focused on men. We expected to replicate previous studies where aggression, especially physical aggression, was associated with CAG repeats in the *AR* gene, i.e., the lower the CAG repeats, the higher the aggression [30]. In addition, we also expected a positive relationship between aggression and social status, since it has been described that the use of direct aggression and especially physical aggression is a common mechanism to increase social status through contest competition in men [18]. However, we failed to find these relationships. CAG repeats and aggression have been consistently associated in several populations [30,31,35]. In this sense, studies with robust experimental designs have demonstrated that this relationship between aggression and polymorphism in the *AR* gene can be observed when relevant aspects of personality, such as self-control, are included in the analysis as mediator variables [31]. Therefore, our results support the notion that the inclusion of personality mediators could be necessary to clarify this relationship, at least for a Western population. Another surprising result was the lack of association between aggression and self-perceived social status. We expected this relationship for physical aggression since this is a typical behavior used in contest competition for status in men [18]. Previous research has noted that increasing status through dominance is a less perdurable strategy than using prestige [55,56]. It could be especially significant in social groups where aggression or coercion is highly penalized, and intergroup conflict is attenuated or not present, such as our sample, which is composed of university students from a Western culture (i.e., a WEIRD sample). In addition, we have applied self-assessed questionnaires of social status without analyzing third ratings of social status in natural groups, which also could affect the relationship with dominance, since this can be appreciated within groups, e.g., [57]. Therefore, we cannot discard a relationship between aggression and self-perceived social status in men; however, more studies assessing prestige and manipulation of groups are needed to support this relationship. In addition, as our study was cross-sectional, individuals of low social status could be employing aggression to gain status and individuals of high social status to maintain their social position. That could blur the relationship between the use of aggression and social status. In fact, we found that verbal aggression was negatively related to social status, suggesting that low-status individuals may be using verbal aggression to improve their position.

Our second prediction was centered on women. We expected a negative relationship between CAG repeats with aggression and a positive association of hostility with status. However, we obtained null results. Studies about these relationships are scarce in the field. However, there is evidence about the effect of testosterone on aggressive behavior in women [27,28] (for a negative effect, see [58]) and about the use of indirect forms of aggression to compete for status [18], justifying our expectations. This lack of relationships could arise for the same reasons in women as for men, evidencing the need to apply a more complex design including personality traits as mediators and experimental manipulations to investigate causal relationships between the use of aggression and changes in social status. Accordingly, we propose the need to include women in future studies because current evidence cannot support the notion that polymorphism of the *AR* gene is not relevant to understanding women’s aggressive behavior or to deny a relationship between hostility and social status.

One of the study’s main limitations rests in the lack of psychological measures that can help us appreciate the relationship between the *AR* gene and aggression. For example, a study performed in the Hadza and Datoga tribes of Tanzania [30] found that the mating strategies of individuals (monogamous versus polygamous) affect levels of aggression. It suggests that controlling for sociosexual attitudes (a psychological proxy of individual mating strategy) may be essential to reveal the relationship between the number of CAG repeats and aggression. In studies performed with occidental populations, the inclusion of sub-clinical traits associated with self-control or a certain degree of sadism is also relevant to find the mentioned relationship, especially in studies not focused on prison samples [31]. In any case, we expected that aggression could mediate the relationship between the number of CAG repeats and social status, but this does not occur. We think that future studies need to include a wide range of psychological and sociosexual aspects of human behavior to obtain a clear notion of the effect (if it exists) of candidate genes over human behavior, especially aggression. Another relevant limitation of our study can be observed from the participants. We have worked with a WEIRD sample of university students, which generates selection biases because of a lack of diversity that could inhibit the robustness of the link between aggression, social status, and CAG repeats. There is an open debate around this topic and a general effort to understand the extent to which biased samples affect behavioral studies [59] and avoid their recurrent use. It must be considered that college students tend to proceed from a similar socioeconomic background. Although the MacArthur Scale was used and the study performed in a state university, the sample may not be adequate to generalize the interpretation of the results. We expect to expand this study to other populations to replicate our results, suggesting that polymorphism in CAG repeats in the *AR* gene might be irrelevant to understanding human aggression.

In conclusion, our study was centered on analyzing the relationship between the polymorphism of CAG repeats of the *AR* gene with aggression and self-perceived social status in young men and women. Obtaining negative results for both sexes opens a discussion about the inclusion of other psychosocial determinants to study these relationships and indicates that both sexes must be included in these studies. This is mainly because there is a lack of studies involving women, and such scarcity does not align with the available data regarding the effect of testosterone on female behavior. In addition, these results indicate the complexity of aggression in our species and in other social mammals, where the proximate causes of dominance and aggression are still not fully understood. However, in the last decade, some factors have received strong support. One of the proximate factors is the extrinsic self-reinforcing winner-loser effect, correlating with the hormones testosterone and cortisone, which are responsible for this phenomenon (see ”Biosocial model of status” in Mazur [60]; the Challenge hypothesis in [61,62]; and “Fitness model of testosterone dynamics” in Geniole & Carré, [63]). The subjective evaluation of a resource, which includes motivation, is the second factor (See “The Male Warrior Hypothesis” in [19,64]). Another factor to consider is one’s family (matrilines) and/or one’s network of supporters (e.g., [65,66]). The final factor, which accounts for the majority of dominance and aggression, is the intrinsic Resource Holding Potential or prior attributes (see [67,68], and “formidability” in humans in Durkee et al. [69]). These attributes are physical strength/fighting ability, age, and sex, which are all characteristics that are influenced by genes and hormones. This network of evidence about proximal causes of aggression must be considered in the design of future studies about the possible genetic factors behind the expression of this behavior.

## Figures and Tables

**Figure 1 behavsci-12-00042-f001:**
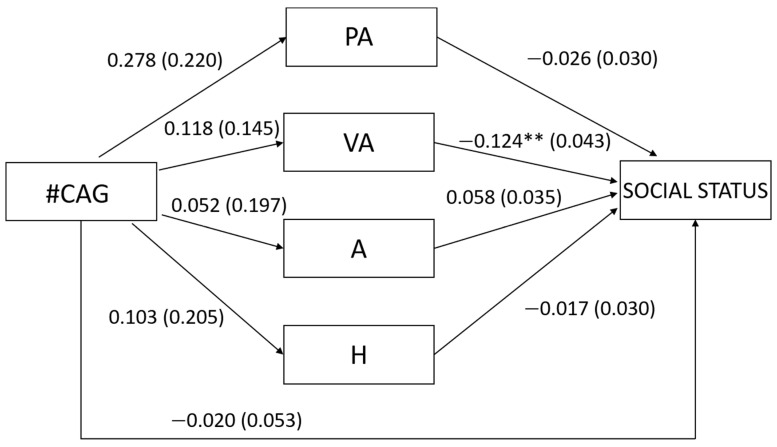
Unstandardized regression coefficients between the number of CAG repeats in the *AR* gene (#CAG) and social status mediated by physical aggression (PA), verbal aggression (VA), anger (A), and hostility (H) in men. Age was controlled in the model but excluded from the figure for clarity. The standard errors are shown in parentheses. ** *p* < 0.01.

**Figure 2 behavsci-12-00042-f002:**
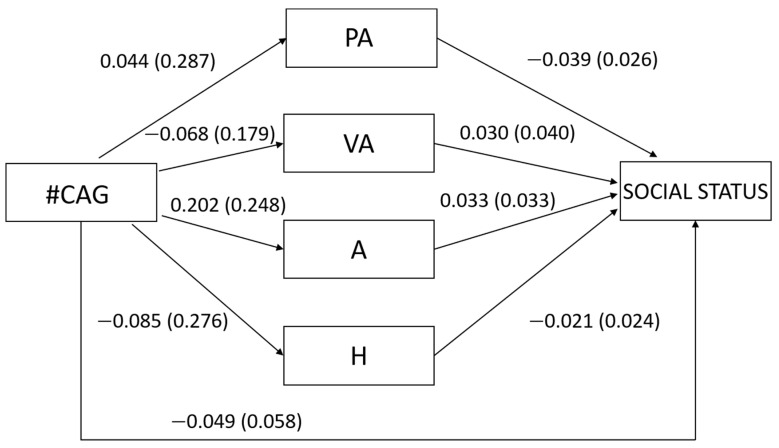
Unstandardized regression coefficients between the number of CAG repeats in the *AR* gene (#CAG) and social status mediated by physical aggression (PA), verbal aggression (VA), anger (A) and hostility (H) in women. Age was controlled in the model but excluded from the figure for clarity. The standard errors are shown in parentheses.

**Table 1 behavsci-12-00042-t001:** Descriptive statistics and *t*-test analysis for the variables employed in this study.

Variables	Men		Women		
	Mean	SD	Mean	SD	t Value (*p*-Value)
Age	20.96	1.58	21.31	2.90	t = −1.042 (*p* = 0.299)
#CAG	23.46	2.75	23.33	2.17	t = 0.367 (*p* = 0.714)
Physical aggression	21.87	5.77	17.43	6.24	t = 5.133 (*p* < 0.001)
Verbal aggression	14.71	3.83	12.78	3.89	t = 3.483 (*p* = 0.001)
Anger	19.69	5.26	18.77	5.42	t = 1.192 (*p* = 0.235)
Hostility	19.41	5.39	19.35	6.01	t = 0.072 (*p* = 0.943)
General aggression	75.67	14.44	68.33	16.54	t = 3.283 (*p* = 0.001)
Social Status	5.86	1.42	6.35	1.25	t = −2.568 (*p* = 0.001)

Note: #CAG: Number of CAG repeats, SD: Standard deviation.

## Data Availability

Data can be consulted in https://mfr.osf.io/render?url=https://osf.io/cm8qx/?direct%26mode=render%26action=download%26mode=render (Accesed on 11 November 2021).

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
