# Peer review of "The Relationship between Androgen Receptor Gene Polymorphism, Aggression and Social Status in Young Men and Women"

_behavsci, 2022, doi:10.3390/bs12020042_

Round 1
Reviewer 1 Report
Dear authors,
Thank you for the paper submitted entitled: 'The relationship between androgen receptor gene polymorphism, aggression, and social status in young men and women', covering an interesting topic. I think this paper tries to predict the relationship among certain forms of the androgen receptor, aggression, and gaining social status.
There is an important issue about the format of references. Please check the journal recommendations because all references do not follow the ACS style guide. Once your manuscript reaches the revision stage, it is mandatory to format the manuscript according to the journal guidelines.This circumstance is inexcusable in a submission to a peer-reviewed journal.
But the main concern I have in this paper is the sample selection. The sample was not randomly selected. The inclusion criterion as volunteers may invalidate the results of the study.
The manuscript should be strongly improved, as I detail in the following specific areas:
Neither in the title nor the abstract, the study design is not indicated.
Introduction:
Lines 31-38/ Lines 48-60 /Lines 66-69: These lines describe the same phenomena: The differences between men and women in terms of the use of aggression to gain social status.
Lines 42-44: As a suggestion, this paragraph should be placed at the end of the Introduction as a justification of the study.
Lines 72-74: A reference needs to be added.
Line 75: testosterone is produced by the ovarian gland in women as well.
Lines 85-87: A reference needs to be added.
Lines 117-122: This paragraph should be placed in the Methods section.
Methods:
The sample was not randomly selected and the sample size was not calculated. The inclusion criterion as volunteers is responsible for a great bias. This is an important limitation of the study that is not described at the end of the article. Perhaps this limitation can invalidate the study. Also, authors compare their sample to WEIRD samples (White, Educated, Industrialized, Rich, and Democratic), therefore, there is a lack of diversity in terms of sociodemographic factors (age, educational level and employment status are the same: University students) that leads to an inconclusive result.
When was the study carried out?
Lines 137-139: Mean age and SD are data that should be placed in the Results section.
Results:
Line 202: RESULTS should be changed into Results.
A sociodemographic table should be included.
Table 1: Should be improved and the first column should be reformulated and better described.
Figure 1 and 2 should be reformulated and better described.
Line 214: The yellow character should be removed.
Discussion:
Lines 249-254: This paragraph should be placed in the Results section.
Limitations are not well explained:
Selection bias
Design bias: The research personal beliefs influence the choice of research question.
Volunteer bias: May produce results or conclusions that differs systematically from the truth, arising where volunteers from a specified sample may exhibit exposures or outcomes that may differ from non-volunteers.
The study is cross-sectional, and it is not possible to establish inferences of causal relationships between variables.
Conclusion:
A Conclusion section should be included but there is no solid conclusion in this paper. Maybe because the sample is not well selected.
References:
Please check the journal recommendations because all references do not follow the ACS style guide.
The references are not updated. 36 out of 56 are older than five years and 9 of them are from the previous century.
Reviewer 2 Report
The manuscript, titled "The relationship between androgen receptor gene polymorphism, aggression, and social status in young men and women," captured my interest. I found it to be quite intuitive, and I became intrigued by the hypotheses put forth in the offer. In general, I have only a few suggestions for authors to take into consideration.
- I would suggest taking into consideration the fluency of the language since some of the manuscript is poor and needs to be improved. Lines 93-97, for example, are extremely confusing and should be rewritten.
- The lines 184-185 refers "Three researchers obtained the measurements independently, and the samples in which there was no coincidence between them were discarded," it. The authors must clarify what the coincidence parameter has been which was used to discard the samples and why it was used?
- Taking into consideration the fact that human social status is highly hierarchical and linear in nature. In what way did the various social contexts in which all of the subjective case study participants self-correlated themselves was taken into consideration during analysis?
- In mammals, the proximate causes of dominance and aggression are still not fully understood. One of the proximate factors is the extrinsic self-reinforcing winner-loser effect correlating with the hormones testosterone and cortisone responsible for this phenomenon. The subjective evaluation of a resource, which includes motivation, is the second factor. Another factor to consider is one's family (matrilines) and/or one's network of supporters. The final factor, which accounts for the majority of the dominance and aggression, is the intrinsic Resource Holding Potential or prior attributes . These attributes are physical strength/fighting ability, age, and sex which are all characteristics that are influenced by genes and hormones. These lines should be included somewhere in the discussion or conclusion section of the paper in order to thoroughly explain the authors' findings.
4. It is recommended that authors separate the limitations of the study and conclusion sections into separate section headings. In addition, they are asked to make suggestions for a possible mechanistic road lines possible to explore the limitations related to study in the future.
Round 2
Reviewer 1 Report
Dear authors,
Thank you for taking the time to address comments on the manuscript entitled: "The relationship between androgen receptor gene polymorphism, aggression and social status in young men and women".
The manuscript has been much improved,but there are still some issues to clarify:
- The manuscript is clear, but it is irrelevant for the field because the sample selection is not well selected. I know that this issue cannot be solved, but this circumstance is inadmissible in a peer review journal.
- The cited references are mostly within before the last 5 years (only around 20 out of 68 are within the last 5 years)
- The experimental design is not appropriate to test the hypothesis because sample selection is not appropriate. The selection of a WEIRD sample (although the MacArthur Scale was used) with young college students in which the high educational level places them in almost the same socioeconomic status (including education and occupational status: similar as they are university students) and therefore this circumstance invalidates the research. Furthermore, the Spanish adaptation of the BPAQ is indicated for pre-adolescents and adolescents, and the mean age of this study is 20.96 years old.
- Another limitation to add is the lack of generalizability (external validity and applicability) of the findings.
Reviewer 2 Report
The article got improved based on the reviewers comments!
Author Response
Dear Reviewer,
We appreciate all the previously received suggestions that improved the quality of the paper.
Sincerely yours,
Miguel Pita
Round 3
Reviewer 1 Report
Dear authors,
Thanks again for your comments that enrich the manuscript. I still think that the selection of the sample is not accurate. Using a WEIRD sample is a big limitation, as it is said in this article:
Hanel PH, Vione KC. Do Student Samples Provide an Accurate Estimate of the General Public?. PLoS One. 2016;11(12):e0168354. Published 2016 Dec 21. doi:10.1371/journal.pone.0168354
It is clear that, in general research, there is a publication bias and it is totally accepted that, although the results do not suit the initial hypothesis, the research should be published. But in this case the results may have been concluded in a different way because of the selection sample. Therefore, I encourage the authors to develop this interesting research with a proper sample.
Hence, we have two facts in the paper that must be taken into account (acknowledged by the authors):
-The results are obtained from a particular WEIRD population, which may produce a certain degree of bias.
-The lack of generalizability (external validity and applicability) of the findings.
So, I consider that this paper should be rejected.
